# Enhanced 3D Pose Estimation in Multi-Person, Multi-View Scenarios through Unsupervised Domain Adaptation with Dropout Discriminator

**DOI:** 10.3390/s23208406

**Published:** 2023-10-12

**Authors:** Junli Deng, Haoyuan Yao, Ping Shi

**Affiliations:** School of Information and Communication Engineering, Communication University of China, Beijing 100024, China; dengjunliok@cuc.edu.cn (J.D.); tinsleyyao@cuc.edu.cn (H.Y.)

**Keywords:** domain adaptation, 3D pose estimation, transfer learning

## Abstract

Data-driven pose estimation methods often assume equal distributions between training and test data. However, in reality, this assumption does not always hold true, leading to significant performance degradation due to distribution mismatches. In this study, our objective is to enhance the cross-domain robustness of multi-view, multi-person 3D pose estimation. We tackle the domain shift challenge through three key approaches: (1) A domain adaptation component is introduced to improve estimation accuracy for specific target domains. (2) By incorporating a dropout mechanism, we train a more reliable model tailored to the target domain. (3) Transferable Parameter Learning is employed to retain crucial parameters for learning domain-invariant data. The foundation for these approaches lies in the H-divergence theory and the lottery ticket hypothesis, which are realized through adversarial training by learning domain classifiers. Our proposed methodology is evaluated using three datasets: Panoptic, Shelf, and Campus, allowing us to assess its efficacy in addressing domain shifts in multi-view, multi-person pose estimation. Both qualitative and quantitative experiments demonstrate that our algorithm performs well in two different domain shift scenarios.

## 1. Introduction

Due to the numerous real-world applications of 3D multi-view multi-person human pose estimation, such as human-computer interaction [1], virtual and augmented reality [2,3], etc., the field of computer vision has seen significant research in this area [4,5,6,7], which has been driven by deep neural networks and large-scale human-annotated datasets [8,9]. These multi-view pose estimation methods have achieved excellent performance on the benchmark datasets [8,9], but still face challenges because of the wide variation in viewpoints, personal appearance, backgrounds, illumination, image quality, and so on. Due to unavoidable domain shifts, pose estimators developed for one particular domain (i.e., the source domain) may not generalize well to novel testing domains (i.e., the target domains). For example, a 3D pose estimator trained on the Panoptic [9] dataset suffers a severe performance drop when evaluated on the Campus [10] and Shelf [10] datasets. In Figure 1, which shows several datasets used for 3D human pose estimation, we can see a considerable domain shift.

Even if more training data from various domains could solve this problem, it may be impractical due to the complexity of real-world scenarios and the high cost of 3D annotation. As a result, methods for successfully transferring a 3D pose estimator trained on a labeled source domain to a new unlabeled target domain are in high demand.

Even if gathering more training data from diverse domains could solve this issue, it may be impractical due to the complexity of real-world scenarios and the high expense of 3D annotation. For this reason, there is a great need for methods that can successfully transfer a 3D pose estimator trained on a labeled source domain to a new unlabeled target domain.

Our research focuses on domain adaptation for multi-view, multi-person 3D pose estimation with covariate shift. We develop an end-to-end deep learning model called “Domain Adaptive VoxelPose” that is based on the cutting-edge VoxelPose model [11]. An unsupervised domain adaptation case occurs when full supervision is available in the source domain but none in the target domain. As a result, there should be no additional annotation costs in the target domain to obtain better 3D pose estimation.

We add three elements to the VoxelPose model to minimize the domain divergence between two domains in order to correct the domain shift. First, we train a domain classifier [12] using adversarial training to learn domain-invariant, reliable features. Second, we apply dropout to multiple discriminators by missing or dropping out each discriminator’s feedback with a specific probability at the end of each batch. This makes the feature extractor more domain-invariant by requiring its output to satisfy a dynamic ensemble of discriminators as opposed to a singular discriminator. Thirdly, we present Transferable Parameter Learning (TransPar) [13] to eliminate the side effects of domain-specific information and enhance domain-invariant learning. TransPar divides all parameters into two categories: transferable parameters and non-transferable parameters. Consequently, TransPar provides distinct update rules for these two categories of parameters.

In conclusion, the following are the principal contributions of our work: (1) We provide domain adaptation components to reduce domain disparity between the selected target domain and the respective source domain. (2) To train a more domain-invariant feature extractor, we propose a novel method that applies an ensemble of dynamic dropout domain discriminators. (3) We employ Transferable Parameter Learning (TransPar) to reduce the negative effects of domain-specific knowledge throughout the learning process as well as to enhance the retention of domain-independent knowledge. (4) The suggested components are incorporated into the VoxelPose model, and the resulting system is capable of end-to-end training. Our method was evaluated on three datasets, and the results suggested that it could enhance the accuracy of cross-domain multi-view multi-person 3D pose estimation.

The paper is organized as follows. Section 2 discusses related works of the methods, including 3D human pose estimation and unsupervised domain adaptation. Section 3 introduces the main baseline, theory, hypothesis, and datasets used in the paper. Section 4 details the proposed methodology for unsupervised multi-view, multi-person 3D pose estimation. Section 5 verifies the feasibility of the proposed model. Section 6 concludes the paper, points out the limitations of the research, and proposes some future work.

## 2. Related Work

### 2.1. 3D Human Pose Estimation

3D pose estimation from monocular inputs [14,15,16,17,18,19,20,21,22] presents an ill-posed problem, as multiple 3D predictions can correspond to the same 2D projection. Multi-view approaches have been developed to alleviate such projective ambiguity. Some methods aggregate data from multiple cameras, utilizing camera settings to identify matching epipolar lines between various views [23]. Additionally, existing camera parameters can be leveraged to generalize new camera setups not present in the training data [24,25]. However, the complexity increases significantly when approaching multi-person scenarios. For multi-person tasks, current approaches mostly use a multi-stage pipeline, including reconstruction-based [10,26,27,28,29,30], volumetric paradigms [11,31], and regression-based approaches [6]. Using a combination of geometric and appearance signals as well as the cycle-consistency constraint, [26] matches 2D postures across several views. Ref. [11] proposes a two-stage volumetric paradigm that circumvents the cross-view matching problem, thereby decreasing the impact of erroneously created cross-view correspondences. Ref. [6] shows skeleton joints as learnable query embeddings and lets them gradually attend to and reason over the multi-view information from the input images to directly regress the actual 3D joint location.

Nevertheless, those efforts focused on the traditional context without taking domain adaptation into account. In this work, we adopt VoxelPose [11] as our foundational pose estimator and augment its generalization capabilities for multi-view, multi-person 3D pose estimation through domain adaptation.

### 2.2. Unsupervised Domain Adaptation

Deep learning algorithms are impacted by the domain-shift problem [32,33,34], which manifests as networks trained on one distribution of data performing poorly on another distribution. This issue typically arises when models are deployed in circumstances that are marginally different from those in the training set. The data from various domains is aligned so that the resulting models have strong generalization performance, which is how unsupervised domain adaptation approaches are typically used to solve this problem. Existing theoretical works can be broadly classified into two types: distance measurement methods and adversarial learning methods. (1) Method based on distance measurement. By reducing the Maximum Mean Discrepancy [35,36], investigate domain-invariant feature learning. (2) Method based on adversarial learning. Inspired by GANs [37], adversarial learning was used to align feature distributions across several domains on various 2D vision tasks [12,38,39,40,41]. Applying dropout to each discriminator’s feedback [42] allows the generator to satisfy a dynamic ensemble of discriminators rather than a static single discriminator, which motivates us to include dropout mechanisms in our domain adaptation component.

Additionally, [43,44] attempts to translate images to close the domain gap at the pixel level. To lessen the HΔH discrepancy, [45] adopted a two-branch classifier. Refs. [46,47] use curriculum learning [48] and sort cases based on how hard it is for them to realize the local sample-level curriculum. Ref. [49] suggests a way of gradually extending the feature norm to close the domain gap. Ref. [13] separates the parameters of a deep unsupervised domain adaptive network into two sections to reduce the impact of domain-specific information, as opposed to previous approaches that depended on learning domain-invariant feature representations. Pose estimators based on volumetric paradigms suffer from computation-intensive 3D convolutions, including many domain-specific parameters. It is suitable to use this Transferable Parameter Learning technique to better utilize domain-invariant parameters.

On par with the developments on domain adaptation for image recognition tasks, some recent works also aim to address the domain shift in regression tasks [31,50,51,52,53,54].

However, despite extensive research on the multi-view, multi-person 3D pose estimation task [6,26,55,56,57,58,59], there are very few jobs that focus on the domain shift problem of multi-view multi-person 3D pose estimation. We propose a novel adversarial training pipeline for domain-adaptive 3D human pose estimation that achieves superior performance in evaluated settings.

## 3. Preliminaries

### 3.1. VoxelPose

We present a brief overview of the VoxelPose model, which serves as the baseline for our research. VoxelPose projects 2D joint heatmaps from various viewpoints into a voxelized 3D space, enabling the direct detection and prediction of 3D human poses. The process begins with the estimation of 2D heatmaps for each view, encoding the per-pixel likelihood of all joints. Features from all camera views are aggregated in the 3D voxel space and processed by the Cuboid Proposal Network to localize individuals. This projection into a common 3D space results in a more comprehensive feature volume, allowing for an accurate estimation of 3D joint positions. Furthermore, the Pose Regression Network is used to estimate a full 3D pose for each proposal. All camera views’ noisy and incomplete information is warped to a common 3D space to create a feature volume that may be used for 3D estimation.

### 3.2. Fundamental Conceptual Framework

The HΔH-Divergence is an influential construct within unsupervised domain adaptation (UDA). In UDA scenarios, we are presented with a labeled source domain, $\hat{S}$, and an unlabeled target domain, $\hat{T}$. The overarching aim is to develop a hypothesis capable of proficiently predicting within the target domain, notwithstanding the lack of its labels.

The groundwork for understanding this theory lies in the theorem titled “Bound with Disparity” [34]. According to this theorem, given a symmetric loss function *ℓ* that adheres to the triangle inequality, the disparity between any two hypotheses *h* and $h^{\prime }$ on a distribution D can be defined as:(1)$$\epsilon_{D}\left(h,h^{\prime }\right)=E_{(x,y)∼D}\left[\ell \left(h\left(x\right),h^{\prime }\left(x\right)\right)\right]$$

Subsequently, the target risk $\epsilon_{T}\left(h\right)$ can be constrained by:(2)$$\epsilon_{T}\left(h\right)\leq \epsilon_{S}\left(h\right)+\left[\epsilon_{S}\left(h^{*}\right)+\epsilon_{T}\left(h^{*}\right)\right]+\left|\epsilon_{S}\left(h,h^{*}\right)-\epsilon_{T}\left(h,h^{*}\right)\right|$$

Here, $h^{*}$ denotes the optimal joint hypothesis.

Building upon this premise, the essence of the HΔH-Divergence is to provide an upper boundary for the disparity difference. The core merit of this divergence, as highlighted in our work, is its capability to be estimated using finite, unlabeled samples from both the source and target domains. However, the direct computation of this divergence is notably intricate and challenging to optimize. Thus, it is approximated by training a domain discriminator *D* that separates the source and target samples. To accomplish this, we employ a dropout discriminator, which not only prevents mode collapse but also enhances the robustness of our algorithm.

The lottery ticket hypothesis refers to a hypothesis in deep learning that suggests finding sparse subnetworks, known as “winning tickets”, within over-parameterized neural networks. These winning tickets can achieve comparable or even better performance than the original large network when trained in isolation under suitable conditions.

In the context of domain adaptation, the lottery ticket hypothesis can be applied to transfer learning scenarios. By identifying winning tickets or subnetworks that generalize well to both the source and target domains, we can effectively adapt the model from the source domain to the target domain. This can help in mitigating the issue of distribution shift between the two domains and improving the performance of the model on the target domain with limited labeled data. This is also the theoretical basis for transferable parameter learning.

### 3.3. Dataset

As shown in Table 1, our evaluation employs three datasets:

Campus [10]: It is a dataset of three people conversing with one another in the outdoor environment, recorded by three calibrated cameras. To assess the precision of the 3D location of the body parts, we utilize the percentage of correctly estimated parts (3D PCP) [10]. We adjust our pose estimator using the dataset’s unlabeled images as the target domain.

Shelf [10]: The Shelf Dataset includes a scenario of ordinary social interactions. In contrast to Campus, this is a more complex dataset, which consists of four individuals deconstructing a shelf at close range. Around them, there are five calibrated cameras, but each view is severely obstructed. 3D PCP [10] is also used as the assessment metric. This dataset’s unlabeled images serve as our target domain.

CMU Panoptic [9]: This dataset was recorded in a lab setting, which contains multiple people engaging in social activities. With hundreds of cameras, it is able to obtain compelling motion capture findings. It is a sizable dataset of social interactions with numerous and various natural interactions. We use it as our source domain.

## 4. Method: Domain Adaptation for Multi-View Multi-Person 3D Pose Estimation

In this section, we detail our proposed methodology for supervised multi-view multi-person 3D pose estimation. In supervised multi-view multi-person 3D pose estimation, we have *n* labeled samples ${\left\{\left(x_{i},y_{i}\right)\right\}}_{i=1}^{n}$ from $X\times Y^{K}$, where $X\in R^{H\times W\times 3}$ represents the input space, $Y\in R^{3}$ the output space and *K* the number of keypoints for each input. The samples randomly selected from distribution *D* are denoted as $\hat{D}$. The objective is to identify a regressor f∈F that yields the lowest error rate ${\mathrm{err}}_{D}=E_{(x,y)∼D}L\left(f\left(x\right),y\right)$ on *D*, where *L* is a loss function we shall explain later. In unsupervised domain adaptation, there exists a labeled source domain $\hat{P}={\left\{\left(x_{i}^{s},y_{i}^{s}\right)\right\}}_{i=1}^{n}$ and an unlabeled target domain $\hat{Q}={\left\{x_{i}^{t}\right\}}_{i=1}^{m}$. The objective is to minimize ${\mathrm{err}}_{Q}$.

### 4.1. Domain Adaptation Component

In the VoxelPose model, the feature representation refers to the feature map outputs of the base convolutional layers (as depicted by the green parallelogram in Figure 2). Specifically, we train a domain classifier to mitigate the domain distribution discrepancy on feature maps. The domain classifier predicts the domain label for each feature map, which corresponds to input images $I_{i}$ from the source or target domain.

This decision has two advantages: (1) Aligning representations at the image level often reduces the shift caused by variations in the images, such as image style, human body scale, illumination, etc. (2) The batch size is typically quite small while training a pose estimation network due to the utilization of high-resolution input. This approach allows more data to be employed in training the domain classifier.

Let’s represent the domain label of the *i*-th training image by $H_{i}$, where $H_{i}=0$ for the source domain and $H_{i}=1$ for the target domain. The feature map of the *i*-th image after the base 2D convolutional layers is denoted by $F_{i}$. Using the cross entropy loss and denoting the output of the domain classifier $D\left(F_{i}\right)$, the domain adaptation loss can be written as:(3)$$L_{D}=-\sum_{i}\left[H_{i}\log D\left(F_{i}\right)+\left(1-H_{i}\right)\log\left(1-D\left(F_{i}\right)\right)\right]$$

To align the domain distributions, we need to simultaneously optimize the parameters of the base 2D network to maximize the above domain classification loss and minimize the keypoint regression loss. In order to optimize the base network used to maximize the domain classifier, we adopt the gradient reverse layer [12], as opposed to regular gradient descent. To maximize the domain classifier loss, the gradient must first pass through a gradient reverse layer, where its sign is inverted.

### 4.2. Dropout Domain Adaptation Component

In order to force the pose estimator to learn from a dynamic ensemble of discriminators, we propose the integration of adversarial feedback dropout in adversarial networks. The feedback of each discriminator is randomly excluded from the ensemble with a specific probability *d* at the end of each batch. This indicates that the pose estimator considers only the loss of the remaining discriminators when updating its parameters.

Figure 2 illustrates our proposed framework, including our initial modification to the adversarial training loss function *L*, as shown in Equation (Equation 2). In this equation, $\delta_{k}$ is a Bernoulli variable $\left(\delta_{k}∼\mathrm{Bern}\left(1-d\right)\right)$, and $\left\{D_{k}\right\}$ is the set of *K* total discriminators. When $\delta_{k}=1$, with $P\left(\delta_{k}=1\right)=1-d$, the gradients derived from the loss of the supplied discriminator $\left\{D_{k}\right\}$, are exclusively employed to produce the final gradient updates for the pose estimator. Otherwise, this discrimination-related information is ignored:(4)$$L_{D}=-\sum_{i}\sum_{k=1}^{K}\delta_{k}\left[H_{i}\log D_{k}\left(F_{i}\right)+\left(1-H_{i}\right)\log\left(1-D_{k}\left(F_{i}\right)\right)\right]$$

Each discriminator trains independently, i.e., is unaware of the others, since no changes are made to their individual gradient updates. Figure 3 depicts the proposed solution’s algorithm in detail.

### 4.3. Transferable Parameter Learning Component

The Transferable Parameter Learning Component is designed to distinguish between transferable and non-transferable parameters, enabling robust unsupervised domain adaptation. To minimize the model’s ability to retain domain-specific information, distinct updating rules are used for different types of parameters [13]. Consider a parameter, denoted by $w_{i}\left(t\right)\in W_{*}\left(t\right)$, its gradient is $\nabla L_{D}\left(w_{i}\left(t\right)\right)$ at the *t*-th iteration. The identifying criterion is defined by
(5)$$T_{i}\left(t\right)=\left|\nabla L_{D}\left(w_{i}\left(t\right)\right)\times w_{i}\left(t\right)\right|,i\in \left[m_{*}\right]$$
where $m_{*}$ is the parameter number of a module in a deep unsupervised domain adaptation network. If the value of $T_{i}\left(t\right)$ is large, $w_{i}\left(t\right)$ is viewed as a transferable parameter. On the contrary, if the value of $T_{i}\left(t\right)$ is small, e.g., zero or very close to zero, $w_{i}\left(t\right)$ is regarded as an untransferable parameter. It is not important for fitting domain-invariant information. If we update it, it will tend to fit domain-specific information. By utilizing objective function gradients and weight decay on the transferable parameters $W^{\mathrm{tr}}$, a robust positive update is executed.
(6)$$W^{\mathrm{tr}}\left(t+1\right)\to W^{r}\left(t\right)-\eta \left(\frac{\partial \left(L_{D}\left(W^{\mathrm{tr}}\left(t\right)\right)\right)}{\partial W_{*}^{r}\left(t\right)}-\lambda W^{\mathrm{rr}}\left(t\right)\right)$$

Furthermore, the non-transferable parameters $W^{\mathrm{utr}}$, which tend to over-adapt to domain-specific details, are negatively updated.
(7)$$W^{\mathrm{utr}}\left(t+1\right)\to W^{\mathrm{utr}}\left(t\right)-\eta \left(\lambda W_{*}^{\mathrm{utr}}\left(t\right)\right)$$
where η>0 refers to the learning rate and W(t) stands for the set of parameters at the *t*-th iteration. The term λW(t) refers to the common weight decay method, which can avoid overfitting by keeping the parameters from being too big. λ is the weight decay coefficient. This method for domain adaptation is straightforward and independent of other approaches.

### 4.4. Network Overview

Figure 2 offers a detailed portrayal of our Domain Adaptive VoxelPose model, an enhancement of the baseline VoxelPose framework with three added components. Primarily, we have incorporated a domain classifier subsequent to the final 2D base convolution layer. Additionally, we have integrated a dropout mechanism that probabilistically neglects the feedback from each discriminator at the end of every batch. Finally, we apply transferable parameter learning to enforce distinct update rules for transferable and non-transferable parameters. The cumulative loss incurred during the training of our proposed network encompasses both the human pose estimation segment as well as the domain adaptation segment. Algorithm 1 provides a comprehensive overview of our approach, which optimizes two kinds of parameters using distinct update rules. It is important to note that our method preserves the architecture of the existing multi-view, multi-person 3D estimation networks. Consequently, both the time complexity and space complexity remain consistent with those of the original networks.
(8)$$L=L_{\mathrm{hm}}+\lambda \left(L_{D}\right)\;$$

**Algorithm 1:** Voxelpose with dropout discriminator and transfer parameter learning.**Input:** Camera Views of source domain, Camera views of target domain**Output:** 3D human poses for all cuboids

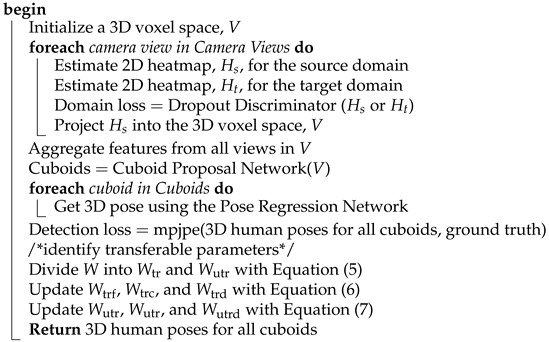



## 5. Experiments

In this section, we verify the feasibility of the proposed Domain Adaptive VoxelPose model. Utilizing the Panoptic [9] dataset as the source domain, the performance of the technique is examined in two distinct scenarios of domain shift: (1) An outdoor environment, where the Campus [10] test dataset captures three individuals interacting outdoors through three calibrated cameras. (2) An indoor social interaction setting, where the Shelf [10] test dataset, which is more complex, features four individuals closely deconstructing a shelf. There are five calibrated cameras surrounding them, but each view has severe occlusion. Due to differences in annotation formats between the source and target domains, a conversion measure is employed to align the model’s output with the target domain annotations (e.g., in a model trained on Panoptic outputs, the position of the nose can be viewed as the head-center position of the Campus dataset).

### 5.1. Experiment Setup

In our experiments, we use the unsupervised domain adaptation approach. The training set is divided into two parts: the source training set, which includes photos and their pose annotations, and the target training set, which only includes unlabeled images. In order to demonstrate the efficacy of the proposed approach, we present not only the conclusive results of our model but also the findings obtained by using each component. This is done for two common domain shift scenarios. We use the original VoxelPose model as a baseline. It was trained using training data from the source domain without taking domain adaptation into account. To assess the accuracy of the estimated 3D poses, we present the Percentage of Correct Parts (PCP3D) across all tests. We use Mean Per Joint Position Error (MPJPE) on the training set.

We conducted our experiments on a computer equipped with an NVIDIA Tesla V100 GPU and implemented our algorithm using PyTorch. For the task of multi-view image feature extraction, we employed a pose estimation model built upon ResNet-50 [11]. The backbone of this model was specifically initialized using weights pre-trained on the COCO dataset. During the model training phase, we utilized the AdamW optimizer [60] with an initial learning rate of 0.001, a batch size of 4, and a total of 20 training epochs. The 2D backbone and the remaining components of the model were trained jointly. Each training batch consisted of two images: one from the source domain and another from the target domain. For training on the Panoptic dataset, which serves as our source domain, we employed three camera views (03, 12, and 23).

### 5.2. Outdoor Environment Experimental Results

With the rapid advancement of 3D human pose estimation, the motion capture (mo-cap) system is emerging as an effective means to augment datasets. However, the Mocap system, originally designed for laboratory use, poses challenges for implementation in natural settings. A discernible visual disparity exists between laboratory data and outdoor scenarios, often leading to a performance gap between models trained in these different environments. Our initial experiment seeks to ascertain the applicability of the proposed method in this context.

**Results:** The results of the different methods are summarized in Table 2. Specifically, using the dropout domain adaptation component alone, we achieve a +5.9% performance gain over VoxelPose. Using the dropout domain adaptation component embedded with transfer parameter learning yields an improvement of 6.9%, validating our hypothesis regarding the necessity of reducing domain shifts. This demonstrates that the domain shift between a lab and an outdoor environment can be effectively reduced by the components we proposed.

As illustrated in Figure 4, the qualitative metrics reveal that our method effectively minimizes false detections. Table 3 demonstrates a significant improvement in performance for joint locations with substantial variations, such as the lower arms and lower legs. We evaluated the generalization performance of our method in comparison to other domain adaptation techniques applied to this scenario, and the results are detailed in Table 4. Based on the results presented in Figure 5, a qualitative comparison is provided between our proposed method and other state-of-the-art algorithms. The results clearly demonstrate that our algorithm exhibits enhanced robustness in cross-domain scenarios.

### 5.3. Indoor Social Interaction Environment Experimental Results

Despite considerable advancements in multi-person 3D human pose estimation, numerous challenging scenarios persist, including obscured keypoints, invisible keypoints, and crowded backgrounds that hinder accurate keypoint localization. For a generalized 3D human position estimation system, precise operation in diverse social interaction environments is vital. This subsection examines the efficacy of 3D pose estimation in the context of group interactions.

**Results:** Our findings, as well as those of other baselines, are included in Table 5. Similar observations apply in the outdoor environment. When all the parts are put together, our complete adaptive Voxelpose model is 2.7% better than the baseline Voxelpose model. In addition, we can see that the improvement generalizes well across different actors, indicating that the proposed method can also reduce domain discrepancies between different individuals.

Figure 6 illustrates the qualitative results of our algorithm, highlighting its capability to not only avoid false detection but also to yield more realistic and natural pose estimation outcomes. In Table 6, notable improvements are evident in joint localizations, particularly for the lower arms and lower legs. In Table 4, we present the results, comparing the generalization performance of our method with other unsupervised domain adaptation techniques applied to this scenario. In Figure 7, we compared our proposed method with other advanced algorithms and found that our approach performs better in cross-domain scenarios. These results indicate that our algorithm is more robust than others.

In Table 7, we evaluate the generalization performance of several state-of-the-art algorithms, including MvP [6], Faster VoxelPose [31], and TesseTrack [57], as detailed in Section 2.2 of the article. Our results demonstrate that in cross-domain scenarios, our method outperforms these other approaches while maintaining the model’s performance on the original dataset.

### 5.4. Ablation Studies and Discussions

#### 5.4.1. Domain Adaptation Component

The training process of the adversarial domain adaptation component is characterized as a zero-sum, non-cooperative contest between the base feature extractor and the domain discriminator. As the domain discriminator learns to distinguish between source and target domain features, the feature extractor simultaneously learns domain-invariant feature representation to confound the domain discriminator, thereby enhancing cross-domain adaptation capability.

Due to the single-adversarial method’s limited distribution alignment ability, the improvement is small and erratic. Mode collapse, as a consequence of overfitting to the feedback of a single discriminator, shows up as difficulties with convergence.

#### 5.4.2. Dropout Domain Adaptation Component

The multi-adversarial domain adaptation method has been empirically validated as an efficacious technique for improving domain adaptation capabilities [41,64]. This method involves dynamically altering the adversarial ensemble at each batch, stimulating the generator to cultivate domain-invariant representations that can deceive the remaining discriminators. The dynamic alteration not only encourages the generator to master domain-invariant representation but also amplifies the probability of successfully misleading any residual discriminators. By aligning the feature representation across diverse feature dimensions, complementary features are learned. This alignment facilitates a more efficient reduction of domain discrepancies with unlabeled target data, thereby bolstering the model’s generalization prowess. The efficacy of the dropout domain adaptive component is further illustrated in Figure 8.

Figure 8 illustrates the relationship between the dropout rate and the generalization ability of the feature representation. This figure emphasizes that selecting an excessively large or small ratio for Parameter *d* can complicate training. By striving to enhance the generalizability of the feature representations created by the base feature extractor, this type of dropout can be seen as a form of regularization. We found that employing any dropout rate within the range (0 < *d* ≤ 1) consistently outperformed a static ensemble of adversaries (*d* = 0). Specifically, utilizing a moderate dropout rate often led to superior results, as previously noted in [42,65].

#### 5.4.3. Transferable Parameter Learning Component

The central concept of the aforementioned components is to acquire transferable feature representations by confusing a domain discriminator in a two-player game, leading to state-of-the-art results in various visual tasks [12,41,64]. Deep Unsupervised Domain Adaptation (UDA) research expects precise feature representations, and insights derived from the source domain can be effectively applied to the target domain. However, during the learning process of domain-invariant features and source hypotheses, unnecessary domain-specific information is inevitably incorporated, hindering generalization to the target domain. The lottery ticket hypothesis [66] reveals that only certain parameters are crucial for generalization. Thus, by eliminating the adverse effects of domain-specific information prior to testing, deep UDA networks can become more resilient and adaptable.

Voxelpose utilizes a 3D-CNN to estimate the 3D locations of the body joints based on the feature volume. This method suffers from a large number of domain-specific parameters. We believe that only partial “transferable parameters” are essential for learning domain-invariant information and generalizing well in UDA; on the other hand, “untransferable parameters” tend to suit domain-specific information and rarely generalize.

In order to lessen the negative impacts of domain-specific knowledge during the learning process, we introduce Transferable Parameter Learning (TransPar) into our network, providing unique update rules for these two categories of parameters.

While Table 2 and Table 5 have shown the benefits of the introduced Learning Transferable Parameters, Figure 9 demonstrates that the performance achieved by medium-to-high ratios is relatively important.

## 6. Conclusions and Outlook

In this research, we present the Domain Adaptive VoxelPose model, which is an efficient method for cross-domain multi-view multi-person 3D pose estimation. Without needing any extra labeled data, one can build a robust pose estimator for a new domain using our method. Our strategy is based on the state-of-the-art VoxelPose model. We introduced a domain adaptation component and the dropout mechanism for our network based on the theoretical analysis that we did for cross-domain pose estimation. A collection of dropout discriminators is used to learn a robust model for the domain. We also introduced transfer parameter learning into our network, which used distinct updating rules for two types of parameters. These components are meant to alleviate the performance drop that is caused by domain shifting. Our methodology is validated on several different domain shift scenarios, and the adaptive method relatively outperforms the baseline VoxelPose method. This demonstrates the approach’s efficiency for cross-domain, multi-view, multi-person 3D pose estimation. In summary, our approach offers industrial advantages by strengthening the robustness of multi-view, multi-person estimation models in real-world conditions, minimizing errors and false positives, increasing operational efficiency, enhancing safety, and alleviating the burden of manual data labeling. However, it’s important to address some inherent limitations in the methodology that could potentially affect its precision and scalability.

### 6.1. Limitations

Primarily, our model confronts the challenge of quantization errors during the transition from 2D to 3D representations. This issue is particularly crucial as it may lead to discernible inaccuracies, undermining the algorithm’s efficacy. Secondly, the computational expenses for training our algorithm escalate with an increasing number of views. This scalability issue limits its applicability in contexts requiring large-scale, high-throughput processing. Moreover, our model currently lacks the incorporation of spatial-temporal data, despite the inherent correlation of human body postures over time. This limitation is especially poignant when the model needs to discern closely intertwined joints or intricate inter-human interactions.

### 6.2. Future Directions

The limitations of the current model naturally guide our future research. One promising direction is the development of an end-to-end framework aiming to mitigate the cumulative effects of quantization errors. Furthermore, we recognize the urgent need for the seamless integration of spatial-temporal data to improve the model’s precision in capturing human interactions and movements over time.

## Figures and Tables

**Figure 1 sensors-23-08406-f001:**
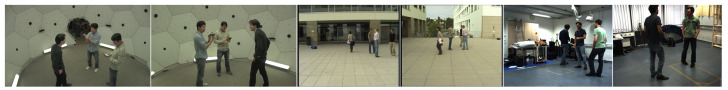
Depiction of various datasets utilized for multi-view, multi-person 3D pose estimation. Image examples are sourced from Panoptic [9], Campus [10], and Shelf [10], respectively. While all datasets feature scenes with clean backgrounds, they differ in aspects such as clothing, resolution, lighting, body size, and more. These visual disparities among the datasets complicate the task of applying pose estimation models across different domains.

**Figure 2 sensors-23-08406-f002:**
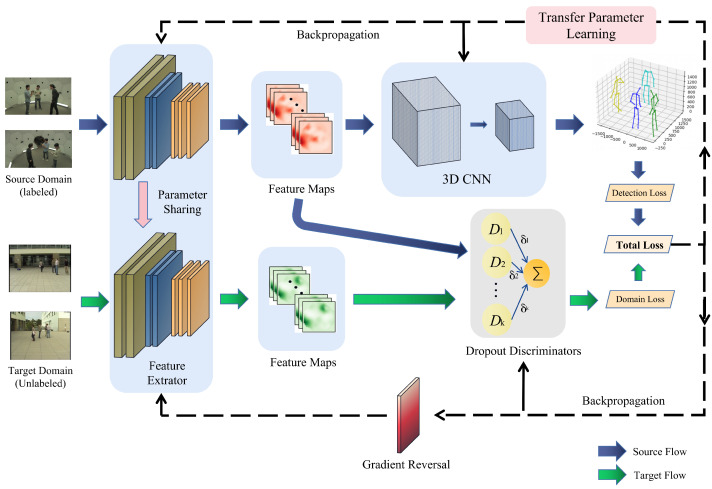
An overview of our Domain Adaptive VoxelPose model. An adversarial training method is used to train the domain classifier. The selection of certain discriminators is determined by a probability $\delta_{k}$. The network performs a robust positive update for the transferable parameters and performs a negative update for the untransferable parameters.

**Figure 3 sensors-23-08406-f003:**
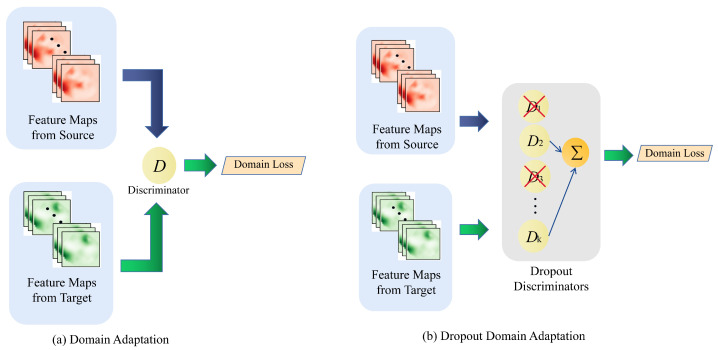
The original adversarial framework (**a**) is extended to incorporate multiple adversaries. In this enhancement, certain discriminators are probabilistically omitted (**b**), resulting in only a random subset of feedback (depicted by the arrows) being utilized by the feature extractor at the end of each batch.

**Figure 4 sensors-23-08406-f004:**
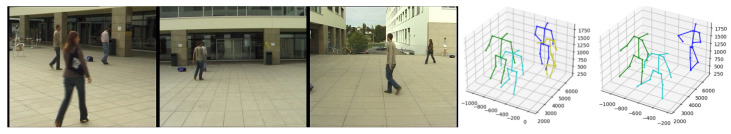
Estimated 3D poses and their corresponding images in an outdoor environment (Campus Dataset). Different colors represent different people detected. The penultimate column is the output result of the original voxelpose, which has misestimated the person. The last column shows the estimated 3D poses by our algorithm.

**Figure 5 sensors-23-08406-f005:**
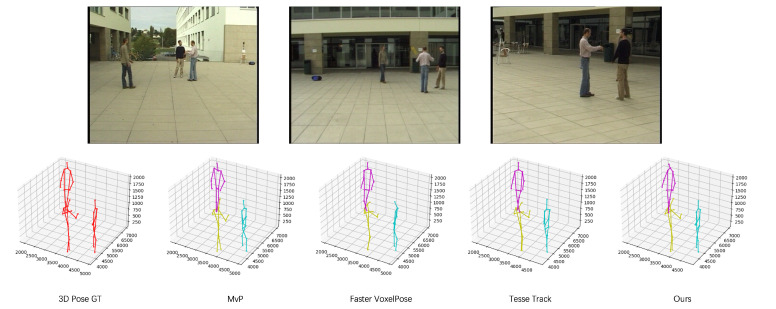
Cross-domain qualitative comparison between our method and other state-of-the-art multi-view multi-person 3D pose estimation algorithms. The evaluated methods were trained on the Panoptic dataset and validated on the Campus dataset. Different colors represent different people detected, with red indicating the ground truth.

**Figure 6 sensors-23-08406-f006:**
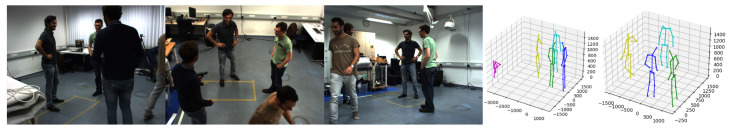
Estimated 3D poses and their corresponding images in an indoor social interaction environment (Shelf Dataset). The penultimate column is the output result of the original voxelpose, which has misestimated person. The last column shows the estimated 3D poses by our algorithm.

**Figure 7 sensors-23-08406-f007:**
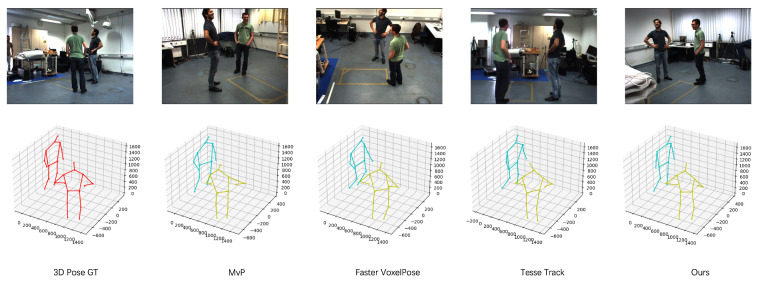
Cross-domain qualitative comparison between our method and other state-of-the-art multi-view multi-person 3D pose estimation algorithms in the Shelf dataset. the evaluated methods were trained on the Panoptic dataset and validated on the Shelf dataset.

**Figure 8 sensors-23-08406-f008:**
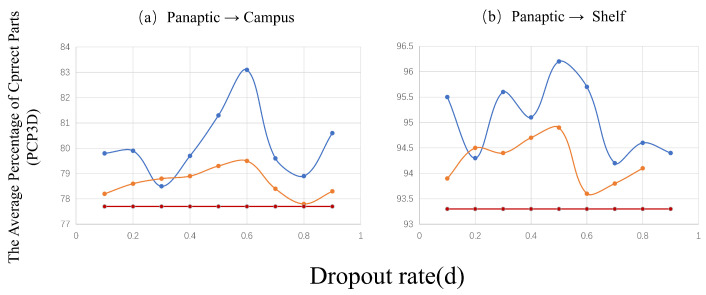
An illustration of the Average Percentage of Correct Parts (PCP3D) on the Campus and Shelf datasets, with the Dropout Rate (d) plotted on the horizontal axis and PCP3D on the vertical axis. The methods are distinguished by color: the red line for the DA baseline method, the yellow line for the dropout DA method, and the blue line for our proposed full method with TransPar.

**Figure 9 sensors-23-08406-f009:**
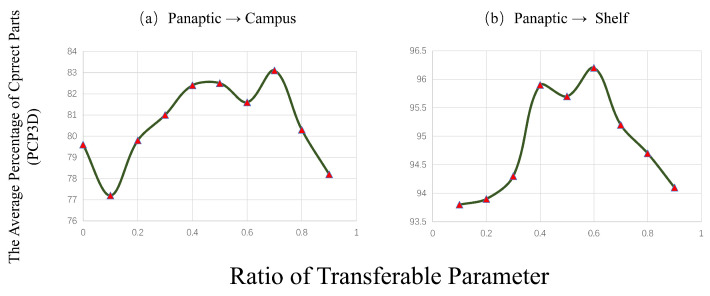
The Average Percentage of Correct Parts (PCP3D), based on Wider Ratios of Transferable Parameters, on Campus and Shelf dataset.

**Table 1 sensors-23-08406-t001:** Overview of the datasets used in our study, emphasizing their characteristics and intended application domains.

Datasets	Duration	Views	Characteristics	Application
Campus	3–4 min	3	3 people on campus grounds.	Target Domain
Shelf	6–7 min	5	4 people disassembling a shelf	Target Domain
Panoptic	60 h	30	Lab-based multi-player interactions	Source Domain

**Table 2 sensors-23-08406-t002:** Quantitative analysis of adaptation results on Campus as a validation set, models are trained on the Panoptic training set. Ours(a) represents the VoxelPose method solely augmented with the domain adaptation module. Ours(b) builds upon the Ours(a) approach by incorporating a dropout mechanism into the discriminator. Ours(c) extends the Ours(a) method by integrating the Transferable Parameter Learning mechanism. Ours(d) represents the optimal approach that amalgamates both the dropout discriminator method and the Transferable Parameter Learning mechanism.

P → C	DA	Dropout DA	TranPar	Actor 1	Actor 2	Actor 3	Average
VoxelPose				78.8	82.8	67.4	76.3
Ours(a)	√			78.6	86.8	79.6	81.7
Ours(b)	√	√		79.3	85.5	78.2	82.2
Ours(c)	√		√	84.3	86.0	78.0	82.8
Ours(d)	√	√	√	85.1	86.3	78.4	83.2

Note: All values are in PCP3D.

**Table 3 sensors-23-08406-t003:** Quantitative analysis of the algorithm’s performance improvement on different keypoints, using Campus as a validation set.

Methods	Voxelpose				Ours(d)			
**Bone Group**	**Actor 1**	**Actor 2**	**Actor 3**	**Average**	**Actor 1**	**Actor 2**	**Actor 3**	**Average**
Head	90.8	68.5	71.4	76.9	91.8	69.7	83.8	81.8
Torso	89.7	95.1	74.7	86.5	90.5	96.2	89.7	92.1
Upper arms	73.5	92.3	71.2	79	82.5	93.1	73.8	83.1
Lower arms	65.9	73.4	58.7	66	78.5	82.3	73.7	78.2
Upper legs	86	92.3	65.1	81.1	85.3	93.4	75.8	84.8
Lower legs	66.9	75.1	63.3	68.4	81.8	83.1	73.3	79.4
Total	78.8	82.8	67.4	76.3	85.1	86.3	78.4	83.2

Note: All values are in PCP3D.

**Table 4 sensors-23-08406-t004:** The Average Percentage of Correct Parts(PCP3D) of Campus and Shelf dataset. Ours(a,b,c,d) represents the different settings of our algorithm, which is explained in Table 2.

Methods	P → C	P → S
VoxelPose [11]	76.3	93.4
VoxelPose-DDC [61]	76.4	90.1
VoxelPose-JAN [62]	81.1	91.2
VoxelPose-DAN [36]	78.7	88.9
VoxelPose-DeepCoral [63]	80.3	87.8
VoxelPose-MMD [61]	74.6	82.5
VoxelPose-RSD [52]	82.2	95.3
Ours(a)	81.7	95.0
Ours(b)	82.8	95.5
Ours(c)	82.5	94.9
Ours(d)	83.2	96.1

Note: All values are in PCP3D.

**Table 5 sensors-23-08406-t005:** Quantitative analysis of adaptation results on Shelf as a validation set, models are trained on the Panoptic training set.

P → S	DA	Dropout DA	TranPar	Actor 1	Actor 2	Actor 3	Average
VoxelPose				93.2	90.5	96.5	93.4
Ours(a)	√			95.1	92.4	97.4	95.0
Ours(b)	√	√		95.2	94.5	96.2	95.6
Ours(c)	√		√	96.0	93.8	97.1	95.1
Ours(d)	√	√	√	96.5	94.1	97.7	96.1

Note: All values are in PCP3D.

**Table 6 sensors-23-08406-t006:** Quantitative analysis of the algorithm’s performance improvement on different keypoints, using Shelf as a validation set.

Methods	Voxelpose				Ours(d)			
**Bone Group**	**Actor 1**	**Actor 2**	**Actor 3**	**Average**	**Actor 1**	**Actor 2**	**Actor 3**	**Average**
Head	78.2	94.6	92.1	88.3	87.2	95.3	94.3	92.3
Torso	98.5	96.1	99	97.9	99.5	96.6	99	98.4
Upper arms	94.3	93.2	96.3	94.6	95.6	93.9	96.8	95.4
Lower arms	93.5	64.9	94	84.1	98.5	81.3	97.2	92.3
Upper legs	96.7	97.4	98.7	97.6	98.3	97.4	98.7	98.1
Lower legs	97.9	96.6	98.5	97.7	99.9	100	100	100
Total	93.2	90.5	96.4	93.4	96.5	94.1	97.7	96.1

Note: All values are in PCP3D.

**Table 7 sensors-23-08406-t007:** Quantitative Comparative Experiments with other State-of-the-Art Methods in Cross-Domain Scenarios. P represents the original performance of Other State-of-the-Art Methods on the panoptic dataset. For our approach, the performance metric of P signifies the retained performance on the original panoptic dataset after undergoing domain adaptation processes. The unit of measurement for P is MPJPE, while others are measured in PCP3D.

Methods	P → C	P → S	P
MvP [6]	80.3	92.6	15.8
Faster Voxelpose [31]	77.1	94.0	18.26
TesseTrack [57]	79.8	92.6	18.7
Ours(a)	81.7	95.0	20.61
Ours(b)	82.8	95.5	19.88
Ours(c)	82.5	94.9	19.56
Ours(d)	83.2	96.1	19.37

## Data Availability

The Shelf and Campus datasets are available at https://campar.in.tum.de/Chair/MultiHumanPose (accessed on 28 September 2022). The Panoptic datasets are available at http://domedb.perception.cs.cmu.edu (accessed on 28 September 2022).

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
