# Peer review of "Enhanced 3D Pose Estimation in Multi-Person, Multi-View Scenarios through Unsupervised Domain Adaptation with Dropout Discriminator"

_sensors, 2023, doi:10.3390/s23208406_

Round 1
Reviewer 1 Report
The paper presents a novel approach to enhance the cross-domain robustness of multi-view multi-person 3D pose estimation. The authors tackle the domain shift challenge through three key approaches: introducing a domain adaptation component, incorporating a dropout mechanism, and employing Transferable Parameter Learning. The proposed methodology is evaluated using three datasets, and the results suggest that it could enhance the accuracy of cross-domain multi-view multi-person 3D pose estimation.
The strengths of the manuscript are the following:
- The paper presents a novel approach to enhance the cross-domain robustness of multi-view multi-person 3D pose estimation.
- The proposed methodology is evaluated using three datasets, which allows for a comprehensive assessment of its efficacy in addressing domain shift.
However, here is a list of improvements:
1) The paper could benefit from a more detailed explanation of the H-divergence theory and the lottery ticket hypothesis, which are the foundation for the proposed approaches.
2) The authors could provide more information on the limitations of their proposed methodology and potential future directions for research.
3) The paper could benefit from a more thorough comparison with existing approaches in the field.
Reviewer 2 Report
Review Comments
The objective of the proposed work is to enhance the cross-domain robustness of multi-view multi-person 3D pose estimation. A domain adaptation component is introduced to improve estimation accuracy for specific target domains. By incorporating a dropout mechanism, we train a more reliable model tailored to the target domain. Transferable Parameter Learning is employed to retain crucial parameters for learning domain-invariant data. However, the following corrections can be considered by the authors to further improve the quality of the manuscript.
I have some major corrections and suggestions below:-
1. Authors must show explain the novel contribution of the work with proper justification of the outcomes. What novelty is established in this work compared to existing works?
2. Organization of the paper can be added at the end of introductions.
3. The abstract need to be improved and the outcome of the work in terms of achieved various other performance calculations must be included in the abstract.
4. Explaining the problem and the gaps in existing literature in a concise but self-contained way (although readers might wish to consult references, they should not be forced to do so)
5. Comparative analysis of various performance parameters with respect to sate of art methods must be discussed. More recent state-of-the-art approaches should be compared; the experiments should use more sizable real-world data sets from public repositories (if any);
6. Comparative analysis with respect to real-time time analysis is missing?
7. Add industrial significance of the proposed approach.
8. Some more Data sets can been utilized. Describing in detail the data set used and what are the expected outcomes- widening the experimental comparison including other data and methods.
9. Comparative analysis of various performance parameters with respect to various other data sets must be discussed. The comparison can be a bit unfair if different data is not used for comparative analysis.
10. How much data should be considered for training and testing for architecture implementation? Details of training and testing data sets must be tabulated.
11. To make the proposed algorithm of this article more readable use pseudo-code.
12. Precision vs. recall curves of the proposed algorithms with respect to data sets must be included.
13. The computational complexity in terms of time and space must be discussed. Also, compare the proposed method in terms of computational complexity?
14. Limitations and Future of the work must be included.
15. Title of the paper need to be rechecked and modified base on proposed work.
16. Implementation platforms with complete specifications of the system must be included.
17. Various visualized results based on proposed work must be added and also compared the results with existing work.
Reviewer 3 Report
Critical remarks:
- Describe the practical construction of the network and the applied transfer learning mechanism. Give the comparison to other solutions.
- Describe the input and output data parameters for transfer learning.
- Table 1 and Table 2 have the same description but somehow they are different.
• Table 1, Table 2, Table 3: give characteristics for positions Ours(a), Ours(b), Ours(c), Ours(d) and other models. Also, describe units.
Based on the given experiments I am still not convinced that the proposed method for multi-view and multi-person 3D pose estimation is efficient. The more experimental evidence is necessary.
Round 2
Reviewer 1 Report
The authors tackled the comments and suggestions successfully.
I believe the manuscript is ready for publication.
Author Response
We are writing to express my sincere gratitude for your positive review of my manuscript, titled "Enhanced 3D Pose Estimation in Multi-Person, Multi-View Scenarios through Unsupervised Domain Adaptation with Dropout Discriminator." Your constructive feedback and insightful comments have significantly enhanced the quality and clarity of the work. Your endorsement is not only an honor but also an invaluable encouragement for our research endeavors.
Thank you once again for your time, effort, and constructive criticism that greatly helped in improving the manuscript. We are looking forward to the possibility of our work being published in Sensors and contributing to the scientific community.
Warm regards,
Junli Deng, Haoyuan Yao, Ping Shi
School of Information and Communication Engineering, Communication University of China,Beijing 100024, China
Reviewer 2 Report
All my concerns and comments has been added successfully
I accept it in current form
Author Response

(The authors gave the same response as above.)

Reviewer 3 Report
Particular remarks:
1. The description of methods in Tables 2, 3, 4 and 5 needs to be more consistent. Give meaningful names to your methods: Ours(a), Ours(b), Ours(c), Ours(d), and describe them. How do these methods differ from the general name Ours in Table 5? Why do the authors sometimes call them Models (Table 4) and sometimes Methods (Table 5)? Add units for numbers in tables. What does column P in Table 5 mean? Why are Ours methods in Table 5 not split?
2. Add quantitative analysis for examples from Fig. 4 and 5.
